# Development of a Responsible Policy Index to Improve Statutory and Self-Regulatory Policies that Protect Children’s Diet and Health in the America’s Region

**DOI:** 10.3390/ijerph17020495

**Published:** 2020-01-13

**Authors:** Sofía Rincón-Gallardo Patiño, Srijith Rajamohan, Kathleen Meaney, Eloise Coupey, Elena Serrano, Valisa E. Hedrick, Fabio da Silva Gomes, Nicholas Polys, Vivica Kraak

**Affiliations:** 1Department of Human Nutrition, Foods, and Exercise, College of Agriculture and Life Sciences, Virginia Polytechnic Institute and State University, Blacksburg, VA 24061, USA; serrano@vt.edu (E.S.); vhedrick@vt.edu (V.E.H.); vivica51@vt.edu (V.K.); 2Advanced Research Computing, Information Technology, Virginia Polytechnic Institute and State University, Blacksburg, VA 24061, USA; srijithr@vt.edu (S.R.); npolys@vt.edu (N.P.); 3School of Visual Arts, College of Architecture and Urban Studies, Virginia Polytechnic Institute and State University, Blacksburg, VA 24061, USA; meaney@vt.edu; 4Department of Marketing, Pamplin College of Business, Virginia Polytechnic Institute and State University, Blacksburg, VA 24061, USA; ecoupey@vt.edu; 5Department of Non-Communicable Diseases and Mental Health, Pan American Health Organization, World Health Organization, Washington, DC 20037, USA; gomesfabio@paho.org

**Keywords:** food and beverage marketing, food policy, nutrition policy, children, integrated marketing communications, government policy, obesity

## Abstract

In 2010, 193 Member States of the World Health Organization (WHO) endorsed World Health Assembly Resolution WHA63.14 to restrict the marketing of food and beverage products high in fat, sugar and salt (HFSS) to children to prevent obesity and non-communicable diseases (NCDs). No study has examined HFSS marketing policies across the WHO regional office countries in the Americas. Between 2018 and 2019, a transdisciplinary team examined policies to restrict HFSS food and beverage product marketing to children to develop a responsible policy index (RESPI) that provides a quality score based on policy characteristics and marketing techniques. After designing the RESPI, we conducted a comprehensive literature review through October 2019 to examine policies in 14 countries in the WHO Americans Region. We categorized policies (*n* = 38) as either self-regulatory or statutory and calculated the RESPI scores, ranked from 0 (lowest) to 10 (highest). Results showed Brazil, Canada, Chile, and Uruguay had the highest RESPI scores associated with statutory policies that restricted point of sale, cartoon, licensed media characters and celebrities; and HFSS products in schools and child care settings, and broadcast and print media. Policymakers can use the RESPI tool to evaluate marketing policies within and across geopolitical boundaries to protect children’s diet and health.

## 1. Introduction

Food and beverage products high in fat, sugar and salt (HFSS) are a major risk factor for obesity and diet-related non-communicable diseases (NCD) [1,2,3,4,5]. The globalization of the food supply has fostered the widespread marketing of processed HFSS food and beverage products to children worldwide. Robust evidence shows that food and beverage manufacturers, restaurants, food retailers, entertainment and media companies use integrated marketing communications (IMCs) to influence young people’s brand awareness, loyalty, attitudes, preferences, purchasing and consumption behaviors that contribute to poor diet quality and health risks [6,7,8]. This influence operates directly through the environment to affect the conditions in which the population live, which represent the social determinants of health. Marketing practices have been recognized as a commercial determinant of health due the strategies, approaches, and power imbalance that corporations exert [9,10]. The power and reach of IMCs are implemented through strategies, techniques, various media channels and platforms, and settings to create consistent and compelling messages that influence children’s diet and health through a cognitive and behavioral process (Figure 1) [11,12,13,14].

The World Health Organization (WHO) endorsed a set of recommendations in 2010 [5] and released a policy framework in 2012 [11] to support the efforts of Member States to implement comprehensive policies that would restrict the marketing of HFSS food and beverage products to children, birth up to age 18 years. This strategy aimed to improve the complex environmental issues to reduce obesity and diet-related NCD risks. Self-regulatory policies are initiated, led and voluntarily adhered to by private-sector businesses or industry firms often independent from government and civil society input. In contrast, statutory regulations are laws, rules, procedures or voluntary guidelines initiated, recommended, mandated, implemented and enforced by national governments to promote a healthy food environment for children [15]. Since 2010, government and industry have introduced policies in Australia, Canada, Chile, Ireland, Mexico, Norway, Sweden, the United Kingdom and United States (USA) to protect children from unhealthy marketing exposure [16,17].

The WHO Global Action Plan to prevent and control NCDs aims to reduce mortality from obesity and NCD by 25% by 2025. This plan included an indicator that tracks weather Member States have adopted policies to restrict the marketing of HFSS food and beverage products to children [18]. In 2015 and 2017, the WHO found that more countries had adopted self-regulatory policies than statutory policies [19,20]. A 2018 WHO report documented that policies exist in 63 (33%) of 193 Member States that had endorsed the 2010 Resolution WHA63.14 to restrict the marketing of HFSS food and beverage products to children, of which almost half (*n* = 30) are government statutory policies [21]. The WHO European Region has the greatest number of policies, but most actions are voluntarily, whereas the WHO Americas Region has the most government statutory policies [19,20,22].

The evidence shows that a majority of policies within and across countries and regions have reduced product advertisements on television and in school settings [16,17,23,24]. However, the marketing of HFSS food and beverage products worldwide has shifted to other IMC techniques, media channels and platforms and settings. This study selected countries from the Americas region due to the recent enactment of government statutory policies and laws to address unhealthy food marketing to children. No study has examined the content of government statutory and industry self-regulatory policies in the WHO/Pan American Health Organization (PAHO) regional office of the Americas. Moreover, no published study has used big data analysis and data visualization tools to assist policymakers to understand and compare the effectiveness of policies within and across countries. More comprehensive approaches are needed to protect children and youth from HFSS food and beverage marketing [17].

The present research addresses these research gaps by examining the characteristics of various policies to restrict HFSS food and beverage product marketing. First, we used a WHO policy framework and IMC strategy framework to develop a responsible policy index (RESPI) to assess characteristics of HFSS food and beverage product marketing to children. Second, we conducted a comprehensive literature review of literature through October 2019, to examine the policies of a purposive sample of 14 countries in the WHO Regional Office of the Americas (i.e., North, Central, South America and the Caribbean). Third, we categorized the relevant evidence on the policies and calculated the RESPI score to reflect the overall quality of the collective policies. Fourth, we created a web-based platform to use data visualization tools to display the RESPI scores of the policies across the 14 countries. Our research indicated that government policymakers and other non-State actors can use the RESPI tool to evaluate marketing policies across countries and regions, to help Member States more fully implement WHA63.14 to protect children’s diet and health.

## 2. Materials and Methods

A transdisciplinary team science approach integrates researchers who have expertise in different disciplines to collaboratively identify new solutions for complex problems [25]. In 2018, we convened a transdisciplinary research team comprised of five academic faculty and two students with expertise in nutritional sciences, policy, business, marketing, visual arts and information technology, to develop a the RESPI for the marketing of HFSS food and beverage products to children. In the present research, a policy is a law, procedure, regulation, rule or standard that guides how government, business and organizations operate and how citizens live their lives [26].

### 2.1. Responsible Policy Index (RESPI) for the Marketing of HFSS Food and Beverage Product to Children

An index is a numeric way of expressing the difference between two measurements, often resulting in a rank-ordering of relevant measures or entities. An index can be used to draw attention to issues, inform publics and policymakers, and stimulate thinking [27]. Indexes that distill large amounts of information into fewer, pertinent numbers are gaining popularity among researchers and policy advocates to facilitate and improve decision making, and to ensure appropriate communication with stakeholder groups that may on food nutrition knowledge and expertise [28,29].

We developed the RESPI for the marketing of HFSS food and beverage products to children to track and rate policy quality, where quality is defined as the standard of something measured against the degree of excellence [30]. We used two conceptual frameworks to construct and select the components and indicators within the index. The WHO policy framework to implement a set of recommendations on the marketing food and non-alcoholic beverage products to children for the policy characteristics. The Integrated Marketing Communications IMC framework (Figure 1) was used to identify the marketing strategies and techniques covered in the policy design [11,12]. Each policy is assessed through two equally weighted components: (1) policy characteristics, and (2) marketing techniques covered in the policy design.

To select the policy characteristics, we chose the 2012 WHO policy framework [11] for two reasons. First, the WHO provides technical assistance to Member States by providing guidance based on the best available evidence. Second, the WHO policy framework was intended for policymakers in Member States to guide the policy cycle (i.e., policy development, implementation, and monitoring and evaluation) to implement Resolution WHA63.14. The document describes that the policy development process as the first step to determine the policy approach, define the terms and scope of the policy. If the policy has a rationale based on a child-rights approach, the country should use the definition of a child outlined in Article 1 of the UN Convention on the Rights of the Child that defines a child as a person below 18 years of age [31], and which foods to include or exclude (i.e., nutrient criteria) [11]. The policy implementation section recommends to the governments statutory regulations, which requires uniform implementation ensuring full coverage [11]. Lastly, the framework highlighted that the WHA urged Member States to establish an accountability system that included monitoring, evaluation and enforcement (i.e., punish or sanctions to infringements) [11].

The marketing techniques were selected from the IMC framework [13] that describe marketing strategies, techniques, media channels and platforms, and diverse settings (Figure 1). These factors were constructed based on evidence from the marketing literature and combined with the 2012 WHO policy framework [11].

The index provides a single score, that reflects the quality of any given policy, based on calculating the number of points from two components: component 1 (policy characteristics) and component 2 (marketing characteristics). Policies are able to obtain up to five points from each component for an overall score of 10 that indicates the highest quality against the consensus reflected in the WHO policy framework and IMC conceptual model. A score of 0 reflects the lowest quality policy. There are three steps working out the overall score of each policy described below (Table 1).

**Step 1:** Calculate the total points from the component 1 “policy characteristics”

A maximum of five points can be awarded for this component. The total points are the sum of the points from each indicator of this component.

**Step 2:** Calculate the total points from the component 2 “marketing techniques”

A maximum of five points can be awarded for this component.

[[total marketing techniques points] × 100/8] /20.

Exceptions: if the policy covers the following media channel, platform or setting.

Broadcast and print: [[total marketing techniques points] × 100/6] /20.

By default, these settings do not cover the marketing techniques of direct marketing and point of sale.

Outdoors and transportation: [[total marketing techniques points] × 100/4] /20.

By default, these settings do not cover the marketing techniques: direct marketing, product placement, point of sale.

**Step 3:** Calculate the overall RESPI score for each policy on the marketing of HFSS food and beverage products to children.

[total policy characteristics points] + [total marketing technique points].

### 2.2. Data Collection

To describe the policy landscape in countries of the Americas Region, we used a purposive sample of 14 countries out of 35 countries that represent Member States from the Pan American Health Organization (PAHO) which also serves as the WHO Regional Office of the Americas. A purposive sample is a non-probability sample that can be logically representative of a population [32]. The 14 countries were selected based on a broad geographical spread across the region of North, Central, South America and the Caribbean. The countries included: Argentina, Bolivia, Brazil, Canada, Chile, Colombia, Costa Rica, Dominican Republic, Ecuador, El Salvador, Mexico, Peru, USA and Uruguay.

To identify the policies in the 14 countries mentioned above, we used the WHO Global database on the Implementation of Nutrition Action (GINA) [33]. We also conducted a comprehensive review of relevant peer-reviewed and gray literature sources (i.e., websites of governmental ministries, industry alliances and trade associations, legal databases, published papers, and governmental and nongovernmental reports) in English, Portuguese and Spanish. We selected policies with extensive data at no cost and without restriction. The obtained data were triangulated and verified using a cross-checked consultation process against each policy source and the primary documentation from government websites. Identified policies without the primary source documents were excluded.

Information for each identified policy was included, extracted, coded and summarized into an Excel database using key words for each policy component and indicators (Table 2). The evidence was coded twice (October 2018 and 2019) to ensure consistency. Thereafter, a joint consensus coding by data and investigator triangulation with another author (V.K.) was followed. After gathering the information, the transdisciplinary research team met in person to resolve any discrepancies in coding the data.

### 2.3. Data Analysis

We analyzed data from policies that restrict the marketing of HFSS food and beverage products to children from a purposive sample of countries (*n* = 14) from the PAHO region. Descriptive statistical analyses were conducted using frequencies, proportions, means, modes, distributions and standard deviations. After the RESPI scores were calculated for each policy, the transdisciplinary research team developed a web-based platform to depict the results using interactive data visualizations. Data visualization tools help to reduce a large and complex database, as of relevant indexes, into an intuitive and easily understood, interactive web-based display to help decision-makers analyze the existing policy landscape, facilitate comprehension, and identify policy solutions based on relevant evidence [19].

The coded data from the Excel spreadsheets were uploaded to the Open Science Framework cloud to manage the data and create the data visualizations. Data extraction was conducted using a Python programming language, mining and querying framework. The interactive web-based querying and visualization framework was created in Ploty Dash. Using the Dash framework also allows us to use the Ploty package for visualizations efficiently. Processing data was using Python Pandas. We used descriptive statistics based on the STATA statistical software package version 16.0 [34].

For visualizations, bar charts, pie charts, scatterplots, histograms and choropleths (color-coded maps) were automatically generated from the data in an interactive manner based on the user selection. The framework consisted of three sections: the top section has dual panes, with the choropleth displaying metrics of interest as well as allowing for context selection for country-based statistics in the right-hand pane. The right-hand pane in this section allows for the selection of various metrics from a drop-down menu. The middle section consists of comparative visualizations for the countries involved. Boxplots were used to display the policy mean, variance and quantile information whereas an overlaid line chart was used to represent the variation in policies over time for all the countries. The bottom section is a filter-enabled data table that displays the raw information for the user.

## 3. Results

We identified 38 policies to reduce the marketing of HFSS food and beverage products to children in a purposive sample of 14 countries of the WHO’s Region of the Americas (Table 3). Nearly three quarters (68%, *n* = 26) of the policies were implemented after the Resolution WHA63.14 was endorsed in 2010, (Figure 2). The highest RESPI scores (i.e., the highest quality policies) were for statutory regulations in Brazil, Canada, Chile, and Uruguay, (RESPI score = 7 to 8). Of these, Brazil and Uruguay had the most recent policies, both from 2018 as described in Appendix A. Colombia, Costa Rica and the Dominican Republic had the lowest quality policies (RESPI score = 1). Colombia had one of the oldest policies, introduced in 1998, that had not been updated.

### 3.1. Policy Characteristics

Fifteen policies (40%) scored more than 3 out of 5 possible points on this component of the RESPI, and all were statutory regulations. The more prevalent indicator from policy characteristics across countries was statutory regulations (82%), followed by using nutrient criteria (60%), monitoring and/or evaluation (76%), using a child-rights approach (45%). The UN definition of children from birth up to 18 years was used infrequently (18%).

#### 3.1.1. Statutory Policies vs. Self-Regulatory Policies

We found several differences between ongoing government statutory and self-regulatory policies to restrict the marketing of HFSS food and beverage products to children (Table 4). None of the self-regulatory policies used a child-rights approach or the UN definition of children from birth up to 18 years. However, there was a difference for monitoring and evaluation, which was more common for self-regulatory (86%) than statutory (58%) policies. Fewer than one third (28%) of the self-regulatory policies compared with statutory (68%) policies used a nutrient criteria or nutrient profiling model to determine the products allowed to be marketed to children.

The marketing techniques covered by the policies were broad. Overall, the most popular policies addressed point of sale, cartoon mascots, licensed and media characters, celebrities, premium offers, and product design and packaging. However, marketing through mobile devices, digital and social media were covered infrequently by both statutory and self-regulatory policies (10% and 14%, respectively).

#### 3.1.2. Child-Rights Approach

None of the self-regulatory policies contained explicit language to protect children’s right to food and health aligned with the 1989 UN Convention on the Rights of the Child [31], compared with the statutory policies (55%). The lowest quality RESPI scores (range: 1–2) were policies that did not use child-rights language.

#### 3.1.3. Monitoring and Evaluation

We found 52% (*n* = 15) of policies that described both a monitoring or evaluation system to ensure compliance along with sanctions in case of violation; 86% (*n* = 25) included a monitoring or evaluation system; and 66% (*n* = 19) included government sanctions for non-compliance with statutory policies.

Of the policies with the lowest RESPI scores (range: 1–3), only three policies from Colombia, Costa Rica and Mexico included both a monitoring or evaluation and sanction indicators. In contrast, five of the eight policies with the highest RESPI scores that (range: 6–8), included both a monitoring or evaluation and a sanction component. Three-quarters of the policies that contained a monitoring and/or evaluation indicator (73%, *n* = 11) were designed to restrict the marketing of HFSS food and beverage products to children at schools and in child-care settings.

#### 3.1.4. Nutrient Criteria or Nutrient Profiling

Twenty-three (60%) policies indicated using nutrient criteria or nutrient profiling to define which food and beverage products were allowed or restricted for marketing to children. The RESPI scores of the policies that defined nutrient criteria ranged from 1 to 8; however, the majority of these policies were statutory (91%, *n* = 21). These policies mainly restricted the marketing of HFSS food and beverage products to children at schools (74%, *n* = 17) and through point-of-sale in retail locations (70%, *n* = 16). Uruguay [35] and Peru [36] were the only countries that had considered PAHO-recommended nutrient profiling [37] system to determine which food and beverage products are allowed to be marketed to children.

#### 3.1.5. Definition of a Child

Statutory policies were most likely to define a child as below the age of 18 years, in line with the UN definition of a child (23%, *n* = 7). All these policies had a RESPI score that ranged from 5 to 8. One third of the policies (32%, *n* = 12) defined a child as up to 12 years; a quarter of policies (29%, *n* = 11) did not specify the age; and the remaining policies varied the definition of a child between 13 and 17 years old. Policies using the UN definition of a child were generally designed to restrict the marketing of HFSS food and beverage to children in schools up to high school and child care settings (Canada [38,39]), and on the product design and packaging as front-of-pack labeling (Uruguay [35], Ecuador [40], Mexico [41]). These front-of-pack labeling policies did not specify children age; it was implied that the policy protect this population. Brazil [42] and Bolivia [43] were the only two countries with policies that fully specified the protection of children up to 18 years of age.

### 3.2. Marketing Techniques

The most common marketing techniques addressed by the policies were point-of-sale (53%, *n* = 20), followed by restrictions for the use of cartoon mascots, licensed media characters and celebrities, (40%, *n* = 15) (Figure 3). Statutory policies were more likely to restrict point-of-sale (61%, *n* = 19) than self-regulatory policies, which were more likely to restrict the use of cartoon mascots, licensed media characters and celebrities, (57%, *n* = 4). Overall, direct marketing (18%, *n* = 7) and branding (24%, *n* = 9) were the marketing techniques covered the least often by the policies reviewed.

Results shows that policies regulated marketing in various media channels, platforms and diverse settings simultaneously, being the most covered by policies, schools and child care settings (58%, *n* = 22), followed by broadcast and print media (50%, *n* = 19). In contrast, digital and social media, and mobile devices were the least included (11%, *n* = 4; 13%, *n* = 5, respectively) in both type of policies (Figure 4). Policies with the highest RESPI scores ranged from 6 to 8, and were developed to protect children from HFSS food and beverage products at schools and child care settings. In comparison, policies with the lowest RESPI scores ranged from 1 to 2 and were developed to restrict the marketing of HFSS products through broadcast and print media.

## 4. Discussion

This is the first study to provide a regional representation and comprehensive overview of statutory and self-regulatory policies to restrict the marketing of HFSS food and beverage products to children in the WHO Region of the Americas. Previous evidence has provided information on the marketing of food and beverage products within a specific media channel [44,45] and reported progress data in implementing the WHO Recommendations [46].

We developed and tested a RESPI tool to assess the quality of the government and industry policies and compare the scores across policies within the several IMC techniques, channels, media, and platforms that industry uses to influence young consumers. This study yields three major findings. First, higher-quality policies (i.e., higher RESPI score) were statutory, and lower-quality policies were self-regulatory. Second, most of the policies covered advertising and marketing of food and beverage products in schools and child-care settings; as well as through broadcast and print media channels. Third, the most frequently restricted marketing techniques covered by these policies were point-of-sale and the industry’s use of cartoon mascots, licensed media characters, and celebrities.

We identified that across the America’s region, most of the statutory policies were implemented starting in 2010, which suggested that these policies gained momentum over time as food and beverage marketing to children became a more prominent issue highlighted by the WHO [4,5,11,18,19,20,44,45,46], and by other UN system organizations such as UNICEF [47,48]. Over the time period reviewed, we found that industry developed self-regulatory policies before 2010 in Brazil, Colombia, and Mexico and subsequently governments developed a comprehensive statutory policy.

Higher RESPI scores were found for statutory policies, while industry self-regulatory policies had the lowest RESPI scores. For example, the self-regulatory program and voluntary code of conduct to reduce the marketing of HFSS food and beverage products to children called Publicidad de Alimentos y Bebidas Dirigidas al Púbilco Infantil (PABI, by its acronym in Spanish) was put in place by industry in Mexico [49]. The PABI had a RESPI score of 2. By 2014, the Mexican government had developed and implemented a statutory policy [41] that banned the marketing of HFSS food and beverage products not aligned with specific food category thresholds on broadcast channels, which had a RESPI score of 5. By 2019, the Mexican government approved a more comprehensive policy of improved quality (RESPI score = 6), including more detailed criteria about the front-of-package labels, use of cartoon mascots, licensed media characters, celebrities, and use of premium offers [50].

These findings reveal the weak quality of existing self-regulatory policies, which had the lowest RESPI scores, and indicate a shift over time to improved policy quality. The results are aligned with previously published evidence regarding the ineffectiveness of self-regulatory policies to effectively reduce children’s exposure to HFSS food and beverage product marketing for several reasons: the vague or selected definition of a child; unclear nutrient criteria or nutrient profiling guidelines; and a limited scope of specific marketing techniques, media channels, platforms and settings covered by these policies [23,51,52,53].

Generally, the examined regulatory frameworks are primarily focused on schools and child care settings, broadcast and print media, whereas digital and social media, mobile devices, and food retail and restaurants were channels least covered by the policies. Many of the other media channels and platforms, and settings depicted in Figure 3 were included least often in the policies. Our results are consistent with other studies published research [17,54].

Marketing techniques covered varied across policies, with point-of-sale, the use of cartoon mascots, licensed media characters and celebrities, along with premium offers the more popular techniques. The high amount of policies that included point of sale was due to the 22 policies that restricted the marketing of HFSS food and beverage products in school settings. Point-of-sale is a marketing strategy that places items in high-traffic areas such as pay-points, on-shelf displays, and vending machines.

A body of research shows that children are constantly exposed to HFSS food and beverage products marketed to them through multiple IMC media channels, platforms and settings (Figure 1). Therefore, Member States that have endorsed the 2010 Resolution WHA63.14 should design and adopt more comprehensive policies to fill the gaps considering the diverse media channels, platforms and settings where marketing takes place to protect children from all HFSS food and beverage marketing. Without comprehensive strategies to tackle this complex environmental issue, the prevalence of children with obesity across the countries examined in the Americas region is estimated to increase substantially by 2030 [55] as described in Appendix A.

Chilean food marketing policy had one of the highest RESPI scores (RESPI score = 8). This was unsurprising, as the 2015 statutory policy [56] is one of the few that covered marketing techniques (i.e., branding, direct marketing and sponsorship) that were not included in the other countries’ statutory policies. Other published studies also found that the Chilean statutory policy is one of the strongest and most promising to protect children from unhealthy product exposure and reduce their future obesity and NCD risks [17,54].

The results of the present research suggest that high-quality statutory policies are politically feasible in the WHO region of the Americans, and that weak policies can be improved by adding specific elements assessed in the RESPI tool. For example, 13 of the 14 countries (except the USA) have ratified the 1989 UN Convention on the Rights of the Child [31] to protect children’s right to adequate food and a healthy diet. Over half (55%, *n* = 17) of the statutory policies used child-rights language to reflect these Member States’ commitment to implement the principles and articles of the Convention as part of their legal obligation to protect and promote children’s rights. Three additional steps that Member States in the PAHO region could take to strengthen existing statutory policies include: (1) defining the age of children as up to 18 years to align with the UN system definition of a child; (2) adopting effective, evidence-based PAHO-recommended nutrient profiling system [37] to determine which food and beverage products are allowed to be marketed to children; and (3) appointing an empowered body to monitor and evaluate policies that restrict HFSS food and beverage product marketing to children [13]. Lastly, governments could strengthen statutory policies beyond advertising by including all IMC strategies, techniques, channels, platforms and setting addressed [13] in the RESPI tool.

These results can be used to inform government policymaking to identify the weakest components of existing policies; share insights across countries about policies with the highest quality and identify key elements to adopt and improve their policies. Furthermore, the index and visualization system can be used by Member States that plan to adopt, design and implement new policies. This study also provided a panoramic view of the diverse IMC strategies that countries are covering by a single policy or multiple policies. This research suggests that the nutrient criteria and narrow scopes on marketing techniques covered in policies are some of the potential loopholes. The nutrient criteria delimit the products that are allowed to be marketed to children, therefore week standards will result in higher exposure of HFSS food and beverage products as previous research have demonstrated [57,58,59,60]. On the other hand, narrowing the scope to a single marketing technique and/or media channels, settings and platforms leaves room for food and beverage companies to continue to market HFSS food and beverage products to young people. This study may encourage policymakers and other decision-makers to reflect on efforts needed to broadly reduce children’s exposure to HFSS food and beverage marketing and diminish its power and negative influence on young people’s dietary choices and health outcomes. Furthermore, since the RESPI was developed using two conceptual frameworks (i.e., the WHO policy framework to implement a set of recommendations on the marketing of food and non-alcoholic beverage products to children, and the integrated marketing communications framework), this tool could be used in any country or region of the world.

This study had several limitations. The analyses used available and official online sources and documents but did not have access to proprietary industry documents or government documents that were not available online. A second limitation was that the RESPI was developed to assess the quality of policies to restrict the marketing of HFSS food and beverage products. However, the study also assessed policies to promote healthy diets using IMC strategies (i.e., food and beverage standards in schools, front-of-pack nutrition label standards) that Member States have implemented. The RESPI tool does not account for the effectiveness of policies and does not provide extra points for the type of nutrient profiling model used and differentiate between a profiling model developed by industry versus a government-appointed body with authority to monitor and evaluate policies. For example, if a Latin American country enacted a policy to ban the advertisement of all sugary drinks to children up to age 18 years, would not receive maximum RESPI points despite the comprehensiveness of this policy. While this tool may be valuable to reduce children’s obesity risk, obesity is such a complex multi-factorial disease where the environment and political context is important to consider, it is not possible to correlate changes in children’s obesity prevalence based on these policies alone.

Strengths of this study include the novel use of two theoretically grounded conceptual frameworks including the WHO policy framework [11] and IMC framework [13], and official UN definitions used to develop the RESPI. This enabled the exploration and assessment of the quality and comprehensiveness of policies over time and across the 14 countries examined. Furthermore, the transdisciplinary research team leveraged different skills and perspectives to develop a tool capable of capturing the complex components of elements of the food and beverage marketing, as well as policy development and assessment.

Future research should test and adapt this tool to include in the scale points for the product-based criteria (i.e., sugary drink restrictions) and the source and quality of the nutrient criteria or nutrient-profiling model used (i.e., PAHO/WHO nutrient profiling model [37] versus industry models). Further studies could also empirically test the web-based platform and data visualization tools developed for this study among policymakers and a non-scientific audience to understand how different stakeholder groups will interpret and translate the results into policies and advocacy efforts. We recommend that the RESPI tool should be tested across all 35 countries in the WHO Region of the Americas, and potentially other WHO regional offices.

## 5. Conclusions

The RESPI tool for marketing of HFSS food and beverage products to children is a proof of concept and a promising tool that government policymakers, researchers and non-State actors could use to review and rate the quality of the government statutory and self-regulatory policies. This study developed and tested a RESPI tool in 14 countries or the WHO Region of the Americas. The results showed that government statutory policies in Brazil, Canada, Chile, and Uruguay had the highest RESPI scores that restricted point of sale and the use of cartoon mascots, licensed and media characters and celebrities; and HFSS products in schools and child care settings, and broadcast and print media. Although the WHO Americas Region has many existing policies intended to restrict the marketing of HFSS food and beverage products, our results show that children remain exposed to diverse commercial and marketing practices. Because it takes a comprehensive approach to policy characteristics and marketing techniques, our policy assessment tool can abet development, implementation and evaluation of policies to restrict marketing of HFSS food and beverage products to protect children’s diet and health. The web-based platform and data visualization tools can be used to provide valuable metrics and can display trends and data in readily understandable formats that enable benchmarking and tracking progress toward the use and impact of comprehensive and robust policies within and across geopolitical boundaries.

## Figures and Tables

**Figure 1 ijerph-17-00495-f001:**
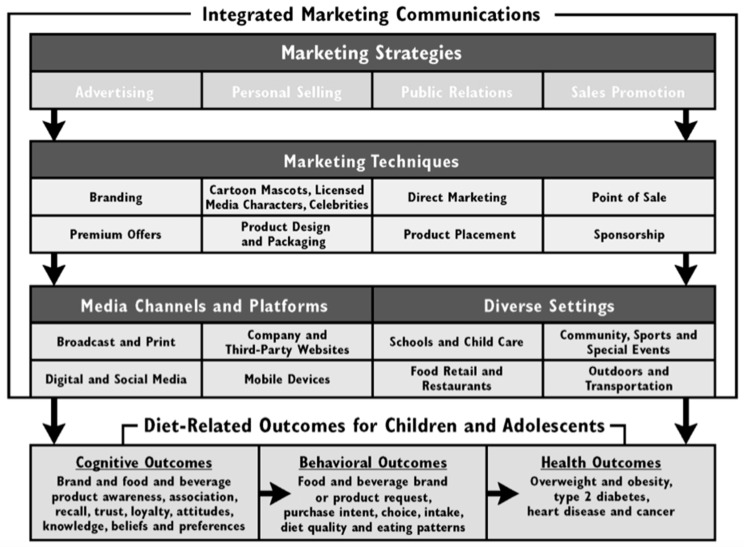
Integrated marketing communications framework of marketing strategies that influence the diet-related outcomes for children and adolescents. Source: Kraak, V; Rincon Gallardo, S; Sacks, G. An accountability evaluation for the International Food & Beverage Alliance’s Global Policy on Marketing Communications to Children to reduce obesity: a narrative review to inform policy. *Obes. Rev.*
**2019**; *20*, 90–106 [13]. Reprinted with permission.

**Figure 2 ijerph-17-00495-f002:**
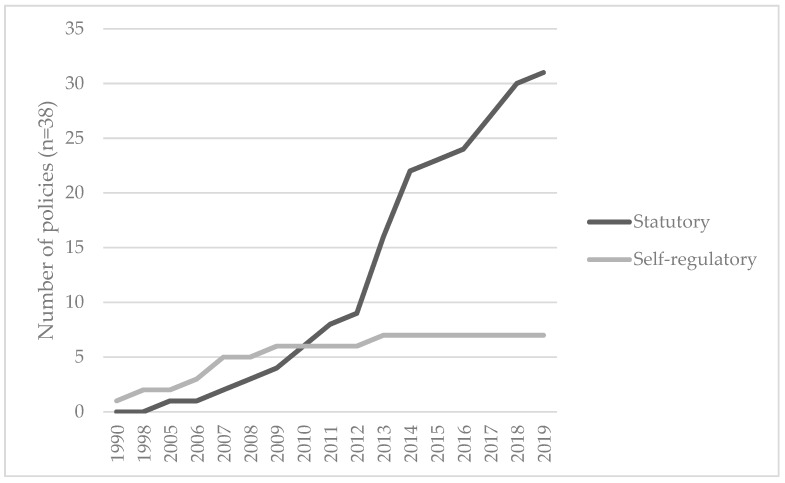
Type of implemented policies that restrict the marketing of food and beverage products in 14 countries from the WHO Americas Region *, 1990–2019. * Data show policy characteristics from a purposive sample of 14 countries of the World Health Organization (WHO) Americas Region, namely, Argentina, Bolivia, Brazil, Canada, Chile, Colombia, Costa Rica, Dominican Republic, Ecuador, El Salvador, Mexico, Peru, Uruguay, and USA.

**Figure 3 ijerph-17-00495-f003:**
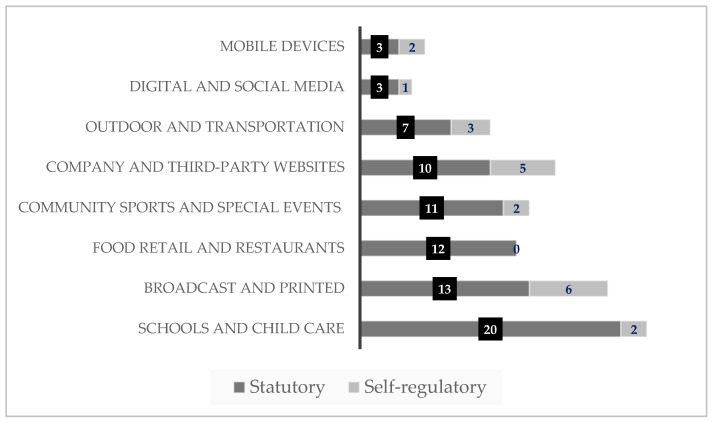
Marketing techniques covered by policies that restrict the marketing of food and beverage products of 14 countries in the Americas Region *. * Data show policy characteristics (*n* = 38) of 14 countries of the World Health Organization (WHO) Americas Region, namely, Argentina, Bolivia, Brazil, Canada, Chile, Colombia, Costa Rica, Dominican Republic, Ecuador, El Salvador, Mexico, Peru, Uruguay, and USA.

**Figure 4 ijerph-17-00495-f004:**
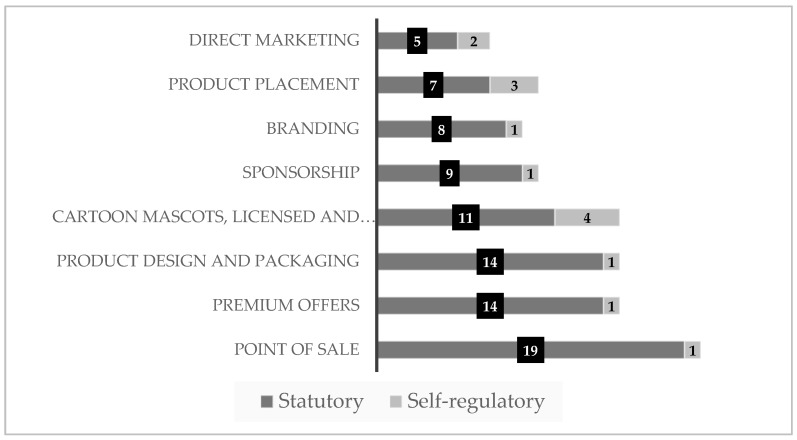
Media channels, platforms, and diverse settings covered by policies that restrict the marketing of food and beverage products of 14 countries in the Americas Region *. * Data show policy characteristics (*n* = 38) of 14 countries of the World Health Organization (WHO) Americas Region, namely, Argentina, Bolivia, Brazil, Canada, Chile, Colombia, Costa Rica, Dominican Republic, Ecuador, El Salvador, Mexico, Peru, Uruguay, and USA.

**Table 1 ijerph-17-00495-t001:** Overview of the responsible policy index (RESPI) methodology.

Indicators	Description	Points
**Component 1: * Policy Characteristics**		
Policy type	Statutory regulatory policy	1
	Industry self-regulatory policy	0
Rights approach	Policy has a right approach	1
	Policy does not have a right approach	0
Monitoring and evaluation	Policy has a monitoring and/or evaluation plan	0.5
	Policy does not have a monitoring and/or evaluation plan	0
	Policy has a sanction component for non-compliance	0.5
	Policy does not have a sanction component for non-compliance	0
Definition of children	Policy use the United Nations age definition of children ***	1
	Does not use the United Nations age definition of children	0
Nutrient criteria	Policy use nutrient criteria to define which food and beverage products are restricted or allowed	1
	Policy does not use nutrient criteria to define which food and beverage products are restricted or allowed	0
Total points (component 1)		0–5
**Component 2: ** Marketing Techniques**		
Branding	Technique that represents a visual name or symbol that legally differentiate and identifies services, products and companies to build economic value, experiences and brand loyalty for the consumer.	
	Included in the policy	1
	Not included	0
Cartoon mascots, licensed media characters, celebrities	Technique that uses third-party, movie-tie-ins, media characters and public figures to promote a product or brand.	
	Included in the policy	1
	Not included	0
Direct marketing	Technique that involves sending a promotional message regarding products directly through direct channels such as mail, email, text messages, catalogues, vouchers, word-to-mouth, samplings, among others	
	Included in the policy	1
	Not included	0
Point of sale	Technique that includes communications activities that take place where products are bought and sold in any sale setting such as on-shelf displays, pay-points, schools and vending machines	
	Included in the policy	1
	Not included	0
Premium offers	Technique that use promotional items such as toys, gifts, coupons, special pricing that can be received for a small fee when redeeming proofs of purchase which come with or on retail products	
	Included in the policy	1
	Not included	0
Product design and packaging	Techniques to attract through image, composition and look of a product that comprise colors, shapes, size and messages used such as cartoon-shaped products, king size, limited edition, and colors	
	Included in the policy	1
	Not included	0
Product placement	Technique that pays for actively seeking to place, promote, or procuring the integration of any message, brand logo	
	Included in the policy	1
	Not included	0
Sponsorship	Technique that involves any form of monetary or in-kind payment that contributes to an activity, event, organization to achieve corporate or marketing-related objectives and directly or indirect promote a product	
	Included in the policy	1
	Not included	0
Total points (component 2)		0–8
**Media Channels, Platforms and Settings**		
Broadcast; and/or company and third-party websites; and/or company and third-party websites; and/or digital and social media; and/or mobile devices; and/or schools and child care; and/or mobile devices; and/or community, sports and special events; and/or food retail restaurants; and/or outdoors and transportation		Yes No

* Based on the World Health Organization (WHO) 2012 framework [11]; ** Based on the integrated marketing communication framework [13]; *** United Nations definition of children: human being below the age of 18 years [31].

**Table 2 ijerph-17-00495-t002:** Search terms used to identify policies from selected countries of the Americas Region for data inclusion, extraction, and coding for the RESPI for the marketing of high in fat, sugar, and salt (HFSS) food and beverage products to children.

Policy Component	Indicator	Key Words
Policy Characteristics	Policy type	mandatory, obligated, statutory
	Rights	rights
	Monitoring, evaluation and accountability	accountable, assess, compliance, enforcement, evaluate, fines, monitor, progress, sanctions, surveillance
	Age	age, years
	Nutrient profiling models	calories, cut off, energy, fat, guidelines, nutrient criteria or profile, sodium, standard, threshold,
Marketing Techniques	Branding	brand, logo, symbol
	Cartoon mascots, licensed media characters, celebrities	actor, actress, brand mascot, cartoon, celebrity, famous, movie-tie-in, media characters, public figure, singer, sport player
	Direct marketing	catalogues, email, face-to-face, mail, sample, text messages, vouchers, word-to-mouth
	Point-of-sale	on-shelf displays, pay-points, vending machines
	Premium offers	coupon, gift, offer, promotion, special price, sale, toy
	Product design and packaging	color, design, edition, package, product shape, size,
	Product placement	place, product, product placement
	Sponsorship	donation, monetary, organization, sponsor
Media Channels, Platforms and Diverse Settings	Broadcast and print	books, comic books, cinema, film, flyers, magazines, newspaper, package, posters, radio, television, video games
	Community, sports and special events	entertainment, event, institution, organization, program, social event, sport event
	Company and third-party websites	digital, electronic, internet, networks, official site, web page
	Digital and social media	digital, electronic, Facebook, internet, Instagram, networks, snapchat, social media, twitter
	Food retail and restaurants	convenience store, farmers market, fast food, market, quick service, retailer, restaurant, supermarket
	Mobile devices	apps, cellphone, laptop, mobile, phone, portables, smartphone, tablet
	Outdoors and transportation	billboards, moving vehicles
	Schools and child care	school property, facility, school district

The key words were developed based on definitions from the following sources: World Health Organization: A Framework for Implementing the Set of Recommendations on the Marketing of Foods and Non-alcoholic Beverages to Children; WHO, 2012 [11]; Healthy Eating Research Recommendations for Responsible Food Marketing to Children; 2015 [14]; Pickton D; Broderick A. Integrated Marketing Communications. Pearson Education: 2005 [12]. Selected countries of the Americas Region: Argentina, Bolivia, Brazil, Canada, Chile, Colombia, Costa Rica, Dominican Republic, Ecuador, El Salvador, Mexico, Peru, Uruguay, and USA.

**Table 3 ijerph-17-00495-t003:** Policies to restrict the marketing of HFSS ** food and beverage products to children in 14 countries of the WHO Americas Region *, 1990–2019.

Country	Policy Document	Year Introduced-Updated	Statutory	RESPI Score
Argentina	Disposición ANMAT No 4980	2005	√	3
Bolivia	Ley No 755 de Promoción de Alimentación Saludable	2016	√	5
Brazil	Resolução -RDC No-24	2010	√	7
Brazil	Resolução 163 Conanda	2014	√	6
Brazil	Código Brasileiro de Autorregulamentação Publicitária del Conselho Nacional de Autorregulamentação Publicitária	1990	×	3
Canada, British Columbia	Guidelines for Food and Beverage Sales in B.C. Schools	2013	√	7
Canada, Quebec	Advertising Directed at Children Under 13 Years of Age: Guide to the Application of Sections 248 and 249 Consumer Protection Act	1980–2012	√	5
Canada, New Brunswick	Healthier Foods and Nutrition in Public Schools	2005–2008	√	5
Canada, all except Quebec	Canadian Children’s Food and Beverage Initiative	2007	×	3
Canada, all except Quebec	The Broadcast Code for Advertising to Children of the Canadian Code of Advertising Standards	2004–2007	×	2
Chile	Ley 20.606 Sobre la Composición de los Alimentos y su Publicidad	2016–2018	√	8
Chile	Código Chileno de Ética Publicitaria	2013	×	2
Colombia	Ley 1355	2009	√	2
Colombia	Decreto 975	2014	√	2
Colombia	Código Colombiano de Autorregulación Publicitaria	1998	×	1
Costa Rica	Decreto No 36910 MEP S Reglamento para el Funcionamiento y Administración del Servicio de Soda en los Centros Educativos Públicos	2012–2013	√	3
Costa Rica	Decreto No 36868-S Reglamento para la autorización y control sanitario de la publicidad de productos de interés sanitario	2015	√	1
Dominican Republic	Ley que Prohíbe la Publicidad Engañosa, Ilícita, Desleal, Subliminal y Discriminatoria en República Dominicana	2011	√	1
Ecuador	Reglamento Sanitario Sustitutivo de Etiquetado de Alimentos Procesados para el Consumo Humano	2014	√	6
Ecuador	Reglamento para la autorización y control de la publicidad y promoción de alimentos procesados	2013	√	5
Ecuador	Reglamento de bares escalares del sistema nacional de educación	2014	√	4
Ecuador	Ley Orgánica de Consumo, Nutrición y Salud Alimentaria	2013	√	4
El Salvador	Acuerdo No 15-0733	2017	√	5
Mexico	Proyecto de Modificación a la Norma Oficial Mexicana NOM-051-SCFI/SSA1-2010	2019	√	6
Mexico	Lineamientos a que se refiere el artículo 25 del Reglamento de Control Sanitario de Productos y Servicios que deberán observar los productores de alimentos y bebidas no alcohólicas preenvasadas para efectos de la información que deberán ostentar en el área frontal de exhibición	2014	√	5
Mexico	Lineamientos generales para el expendio y distribución de alimentos y bebidas preparados y procesados en las escuelas del Sistema Educativo Nacional	2013	√	4
Mexico	Lineamientos por los que se dan a conocer los criterios nutrimentales y de publicidad que deberán observar los anunciantes de alimentos y bebidas no alcohólicas para publicitar sus productos en televisión abierta y restringida, así como en salas de exhibición cinematográfica	2014	√	4
Mexico	Código de Autorregulación de Publicidad de Alimentos y bebidas No Alcohólicas dirigida al Público Infantil	2009	×	2
Peru	Ley de Promoción de la Alimentación Saludable para Niños, Niñas y Adolescentes incorporando el Semáforo Nutricional	2018	√	5
Peru	Ley de Promoción de la Alimentación Saludable para Niños	2013	√	4
Uruguay	Rotulado de los alimentos envasados en ausencia del cliente, librados al consumo en el territorio nacional	2018	√	7
Uruguay	Ley no 19.140 Alimentación Saludable en los Centros de Enseñanza	2013	√	6
USA, Santa Clara County	Ordinance No NS 300–820	2010	√	4
USA, San Francisco	Health Food Incentives Ordinance	2011	√	4
USA, California	Assembly Bill No 841—An act to add Section 49431.9 to the Education Code, relating to pupil nutrition	2017	√	4
USA	Children’s Food and Beverage Advertising Initiative	2006	×	3
USA, Maine	Brand-specific advertising	2007	√	3
USA, California	Legislation-Restricted Marketing of Unhealthy Foods in Schools	2017	√	5

** HFSS: high in fat, sugar and salt; * Data show policy characteristics from a purposive sample of 14 countries of the World Health Organization (WHO) Americas Region, namely, Argentina, Bolivia, Brazil, Canada, Chile, Colombia, Costa Rica, Dominican Republic, Ecuador, El Salvador, Mexico, Peru, Uruguay, and USA. Policy was defined: as a law, procedure, regulation, rule or standard that guides how government, business and organizations operate and how citizens live their lives [26]. Statutory: laws, rules, procedures or voluntary guidelines initiated, recommended, mandated, implemented and enforced by national governments to promote a healthy food environment for children [13]. If the policy was marked with a cross mark it means that is self-regulatory: initiated, led and voluntarily adhered to by private-sector businesses often independent from government and civil society input [13].

**Table 4 ijerph-17-00495-t004:** Comparison of statutory and self-regulated policies that restrict the marketing of HFSS food and beverage products to children of 14 countries in the Americas Region *.

Policy Elements	Statutory (*n* = 31, 82%)	Self-Regulatory (*n* = 7, 18%)	Total (*n* = 38, 100%)
High quality * RESPI scores of 6 and above	8, 26%	0, 0%	8, 21%
Low quality * RESPI scores of 5 and below	23, 74%	7, 100%	30, 79%
Rights approach	17, 55%	0, 0%	17, 45%
Monitoring and evaluation	18, 58%	6, 86%	29, 76%
Nutrient criteria or nutrient profiling	21, 68%	2, 28%	23, 60%
United Nations definition of children (birth up to 18 years)	7, 23%	0, 0%	7, 18%
**Marketing Techniques**			
Branding	8, 26%	1, 14%	9, 24%
Cartoon mascots, licensed and media characters, celebrities	11, 35%	4, 57%	15, 40%
Direct marketing	5, 16%	2, 28%	7, 18%
Point of sale	19, 61%	1, 14%	20, 53%
Premium offers	14, 45%	1, 14%	15, 39%
Product design and packaging	14, 45%	1, 14%	15, 39%
Product placement	7, 23%	3, 43%	10, 26%
Sponsorship	9, 29%	1, 14%	10, 26%
**Media Channels, Platforms and Settings**			
Broadcast and printed	13, 42%	6, 86%	19, 50%
Company and third-party websites	10, 32%	5, 71%	15, 39%
Schools and child care	20, 65%	2, 28%	22, 58%
Community sports and special events	11, 35%	2, 28%	13, 34%
Digital and social media	3, 10%	1, 14%	4, 11%
Mobile devices	3, 10%	2, 28%	5, 13%
Food retail and restaurants	12, 39%	0, 0%	12, 32%
Outdoor and transportation	7, 23%	3, 43%	10, 26%

* Policy characteristics (*n* = 38) that restricted the marketing of high in fat, sugar, and salt (HFSS) food and beverage products to children of 14 countries of the World Health Organization (WHO) Americas Region, namely, Argentina, Bolivia, Brazil, Canada, Chile, Colombia, Costa Rica, Dominican Republic, Ecuador, El Salvador, Mexico, Peru, Uruguay, and USA.

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
