# Peer review of "Development of a Responsible Policy Index to Improve Statutory and Self-Regulatory Policies that Protect Children’s Diet and Health in the America’s Region"

_ijerph, 2020, doi:10.3390/ijerph17020495_

Round 1
Reviewer 1 Report
The paper addresses a very important question in the area of health policy. It uses big data analysis and data visualization tools to assist policymakers in the Americas to understand and compare the effectiveness of policies across countries.
The paper develops an index for measuring the quality of the policy based on score based on policy characteristics and marketing techniques. The paper then proceeds to apply this framework to the Americas. The paper is therefore original, on two counts, in its contribution.
I would like to see more discussion on the development of the index. Why were these factors chosen, were some considered and not included, what are we missing from the measuring kit? How applicable is the measurement tool to other geographic contexts and what other aspects should those researchers consider. Such a discussion would increase the opportunity for replicability in other environments, helping with citations, and also increase the thoroughness of the contribution.
In table 4 – should read ‘rights approach’ (not a right approach)?
Author Response
Dear Reviewer,
Thank you for providing relevant input to the submitted manuscript titled ‘Development of a Responsible Policy Index to improve statutory and self-regulatory policies that protect children’s diet and health in the America’s region’ to the special Issue Marketing of Foods and Beverages: Impact and Potential Solutions for Children and Young People’s Health from the International Journal of Environmental Research and Public Health.
We appreciate the time and effort that you have dedicated to providing your valuable feedback on the manuscript. We are grateful to the reviewers for their insightful comments on the paper. We have been able to incorporate changes to reflect most of the suggestions provided. We have highlighted the changes within the manuscript and attached a document that explains point-by-point the details of the revisions in the manuscript and our responses all of the reviewers' comments.
Sincerely,
Sofia Rincon Gallardo Patino, MSc, RD
PhD student, Human Nutrition, Foods and Exercise (HNFE) program
Virginia Tech, Blacksburg VA

Reviewer 2 Report
Very interesting, important, and timely topic. Overall well written, clear, and highly readable.
In some areas, the authors may want to consider the relationship between commercial determinants of health and social/structural determinants of health, as in some places the language leans a little toward individual victim-blaming. I would recommend removing any emphasis from obesity and keeping the focus on the political environment rather than people's individual behaviours (which are the result of these complex environmental issues).
Minor suggestions stem primarily from the methods section, where clarity is needed:
Line 93: Why list the specific age here? Consider whether this could be broadened to say 'children and youth', or if a specific age is being delineated, indicating why and referencing More information about the RESPI is needed - how were categories chosen, etc. why specifically the WHO region of the Americas? More info on the scoping review method is also needed More clarity needed on data analysis - the web platform, as described looks more like a form of knowledge translation than analysisAuthor Response
Dear Reviewer,
Thank you for providing relevant input to the submitted manuscript titled ‘Development of a Responsible Policy Index to improve statutory and self-regulatory policies that protect children’s diet and health in the America’s region’ to the special Issue Marketing of Foods and Beverages: Impact and Potential Solutions for Children and Young People’s Health from the International Journal of Environmental Research and Public Health.
We appreciate the time and effort that you have dedicated to providing your valuable feedback on the manuscript. We are grateful to the reviewers for their insightful comments on the paper. We have been able to incorporate changes to reflect most of the suggestions provided. We have highlighted the changes within the manuscript and attached a document that explains point-by-point the details of the revisions in the manuscript and our responses all of the reviewers' comments.
Sincerely,
Sofia Rincon Gallardo Patino, MSc, RD
PhD student, Human Nutrition, Foods and Exercise (HNFE) program
Virginia Tech, Blacksburg VA
